**Subject Category:**
Biology (whole organism)

evolution/palaeontology/developmental biology

Early Carboniferous, earliest Serpukhovian (Namurian), adelospondyls, aïstopods, colosteids, 'lepospondyl' polyphyly

**Author for correspondence:**
Jennifer A. Clack
e-mail: jac18@cam.ac.uk

# *Acherontiscus caledoniae*: the earliest heterodont and durophagous tetrapod

Jennifer A. Clack[1], Marcello Ruta[2], Andrew R. Milner[3], John E. A. Marshall[4], Timothy R. Smithson[1] and Keturah Z. Smithson[1]

[1]University Museum of Zoology, Downing Street, Cambridge CB2 3EJ, UK
[2]School of Life Sciences, University of Lincoln, Joseph Banks Laboratories, Green Lane, Lincoln LN6 7DL, UK
[3]Department of Earth Sciences, Natural History Museum, Cromwell Road, London SW7 5BD, UK
[4]School of Ocean and Earth Science, National Oceanography Centre, University of Southampton, Waterfront Campus, European Way, Southampton SO14 3ZH, UK

JAC, 0000-0003-0017-5831; MR, 0000-0002-6151-0704; TRS, 0000-0002-6546-1145

The enigmatic tetrapod *Acherontiscus caledoniae* from the Pendleian stage of the Early Carboniferous shows heterodontous and durophagous teeth, representing the earliest known examples of significant adaptations in tetrapod dental morphology. Tetrapods of the Late Devonian and Early Carboniferous (Mississippian), now known in some depth, are generally conservative in their dentition and body morphologies. Their teeth are simple and uniform, being cone-like and sometimes recurved at the tip. Modifications such as keels occur for the first time in Early Carboniferous Tournaisian tetrapods. *Acherontiscus*, dated as from the Pendleian stage, is notable for being very small with a skull length of about 15 mm, having an elongate vertebral column and being limbless. Cladistic analysis places it close to the Early Carboniferous adelospondyls, aïstopods and colosteids and supports the hypothesis of 'lepospondyl' polyphyly. Heterodonty is associated with a varied diet in tetrapods, while durophagy suggests a diet that includes hard tissue such as chitin or shells. The mid-Carboniferous saw a significant increase in morphological innovation among tetrapods, with an expanded diversity of body forms, skull shapes and dentitions appearing for the first time.

## 1. Introduction

The Early Carboniferous Period (Mississippian) saw the dawn of continental tetrapod diversity. Pentadactylous limbs [1], increased

eye size [2], steep-sided skulls [3] and a wide range of body sizes are found among Tournaisian forms [4]. However, body shape and dental morphologies appear to have remained essentially conservative. By the later Viséan stage, tetrapods had begun to assume more varied body morphologies. Some groups had reduced or lost their limbs and developed elongate vertebral columns [5–7]. The foundations of the tetrapod crown group had been laid [8]. Little attention has so far been paid to dental morphologies, overlooked as conservative. One of the few modifications noted among Tournaisian tetrapods was the appearance of lateral keels on the tooth crowns of a large un-named tetrapod [4]. Here we report the earliest documented example, from the Pendleian (earliest Serpukhovian stage, late Mississippian), of both heterodont and durophagous dental adaptations in the small, limbless and elongate tetrapod, *Acherontiscus caledoniae*. Furthermore, we note the contemporary evolution of three clades of limbless, elongate tetrapods, each with its own specialized dentition [5–7].

*Acherontiscus* was first described by Carroll [9], who recognized its significance, and illustrated but did not discuss its heterodont dentition. More recently, its heterodonty was further revealed by micro-CT scanning [10]. Carroll [9] considered *Acherontiscus* to be a 'lepospondyl', an assemblage of small tetrapods with solid 'holospondylous' vertebral centra and lacking a spiracular or 'otic' notch at the back of the skull. Now recognized as probably a polyphyletic array (e.g. [7]), 'lepospondyls' are split among various Palaeozoic tetrapod groups, leaving the relationships of *Acherontiscus* unresolved. Other small, elongate and limbless tetrapods from the Pendleian include aïstopods and adelospondyls: the relationships of the latter two to *Acherontiscus* have remained little explored.

# 2. Material and methods

## 2.1. Holotype and only specimen: National Museums Scotland (NMS) G 1967.13.1

Purchased by the museum in the late nineteenth century, the specimen of *Acherontiscus caledoniae* consists of a skull about 15 mm long (figure 1) and an elongate, diplospondylous vertebral column, the latter in the natural mould. There are no field or locality data, but recent work has refined the dating to the Pendleian (early Serpukhovian) stage (electronic supplementary material, S1). Although the locality remains uncertain, the specimen is regarded as most likely originating from one of the 'Ironstone' horizons from a colliery in the region of Loanhead, Scotland, probably Burghlee [11]; see also Andrews & Brand [5].

## 2.2. Visualization

Micro-CT at NHM Zeiss versa: Source: 110 kV, 10 W; Camera Binning 1; Exposure: 2 s; Rotation: 180 (Fan); Projections 2501; source distance—50.01 mm; detector distance: 65.95 mm; Filter HE1 (Silicon Dioxide); Lens Objective 0.4×; resolution 14.873 µm; slice dimensions, X 2008, Y 2048, Z2034. Segmentation using Materialise's Interactive Medical Image Control System (MIMICS) Research v18.

### 2.2.1. Microphotography

For figure 1*a*, Nikon D60 fitted with an AF-S DX Micro-Nikkor 40 mm f/2.8G Macro lens. For figure 1*c*, Z-stacks were taken on a Leica S8APO dissecting microscope (Leica Microsystems (UK) Ltd), with a Nikon D5200 camera and Nikon Camera Control Pro 2 using a MacBook Pro computer and rendered into a single focused image with Helicon Focus.

See electronic supplementary material, S2 for the micro-CT scan data and three movies: *Acherontiscus* left lower jaw removed (6.6 MB); *Acherontiscus* left lower jaw (2.5 MB); and *Acherontiscus* skull roof (4.2 MB).

## 2.3. Palynological analysis

Two samples (Ach-1 (0.1 g) and Ach-2 (less than 0.1 g)) of the matrix of *Acherontiscus* were surfaces inspected for contamination or consolidant from curation and conservation. Both appeared to be clean fragments and represented single laminae from the sample. These very small samples were processed using screw-topped Savillex™ PFA digestion vessels and small sieves to preserve the residue and prevent any contamination. Treated uncrushed with 60% HF for 16 h, they failed to disaggregate because of the highly organic matrix. They were then decant-washed clean of HF and subjected to

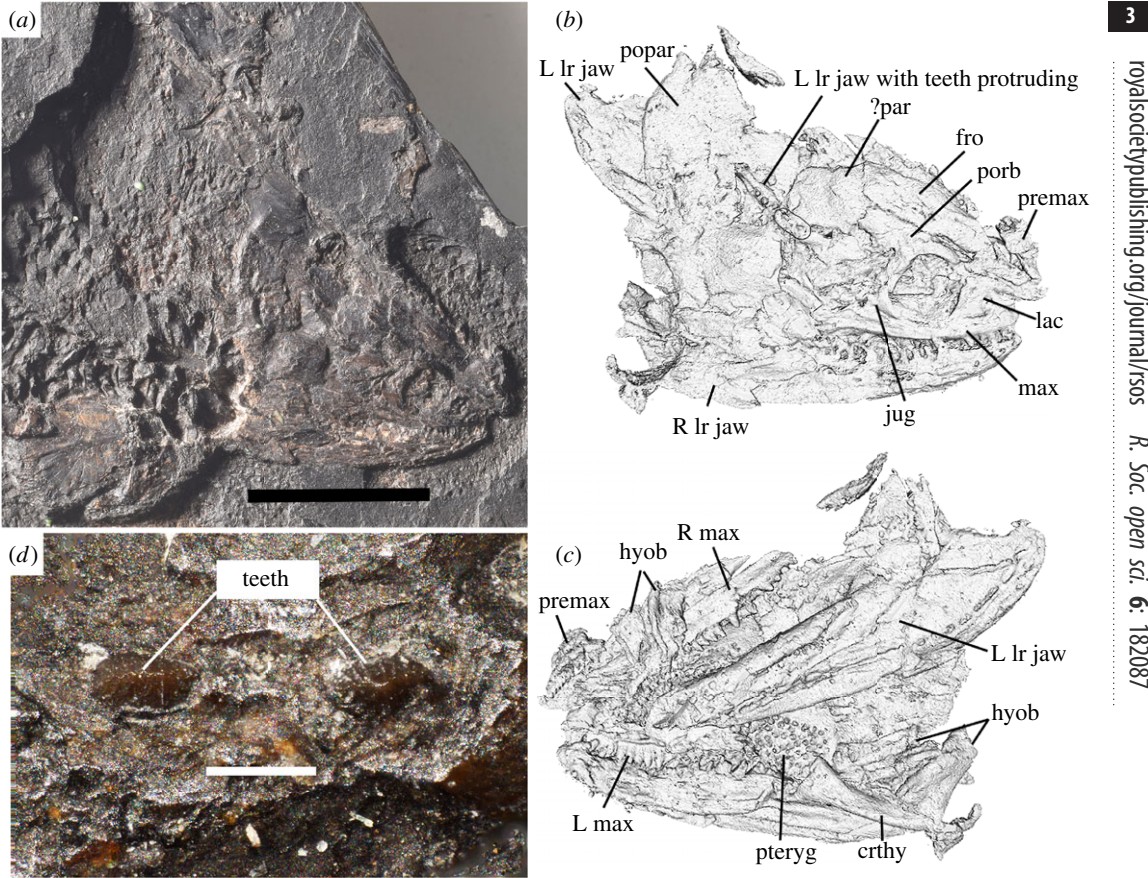

**Figure 1.** *Acherontiscus caledoniae* holotype specimen NMS 1967.13.1. (*a*) Photograph of the skull and anterior postcranium. Scale bar 10 mm. (*b*) View of micro-CT image of the visible surface (approximately dorsal) of the skull. (*d*) Close-up of the crowns of two of the left dentary teeth showing apicobasal ridges. Scale bar 0.25 mm. (*c*) View of micro-CT image of the matrix-embedded (approximately ventral) surface of the skull. Abbreviations: crthy, ceratohyal; fro, frontal; hyob, hyobranchial; jug, jugal; L, left; lac, lacrimal; lr, lower; max, maxilla; ?par, ?parietal; popar, postparietal; porb, postorbital; premax, premaxilla; pteryg, pterygoid; R, right.

15 h in fuming nitric acid. Ach-2 disaggregated readily but Ach-1 failed to dissolve so was crushed down to sub-mm size and returned to fuming nitric acid for a further 15 h, which was successful. The residues were diluted in water and sieved at 15 μm to concentrate the spores. Both samples contained residual minerals including mica and were returned to the digestion vessels for 12 h in 60% HF and resieved at 15 μm before storage. Multiple slides of the very small amount of residue were mounted using Elvacite 2044™ to produce one sparse, poorly preserved but workable palynological assemblage from Ach-1 and a better assemblage from Ach-2 (electronic supplementary material, S1, figures S1 and S2 and table S1).

## 2.4. Phylogenetic analysis

To evaluate the phylogenetic position of *Acherontiscus*, we assembled a new data matrix consisting of 260 characters and 57 taxa, representing a modified and expanded version of the matrix in Clack *et al*. [4]. All 213 characters in that matrix were re-assessed and their scores re-checked for each taxon, and 47 characters were added (electronic supplementary material, S3; new characters marked with asterisks). In order to cover a wider cross section of early tetrapod diversity, in addition to *Acherontiscus*, we added 11 taxa to the Clack *et al*. [4] matrix. Both maximum-parsimony using different character weighting schemes and Bayesian inference analyses were performed. Before parsimony analyses were carried out, the matrix was scrutinized for possible occurrences of 'rogue' taxa [12] that could be safely removed, using the *Claddis* library [13] in the R environment for statistical computing and graphics (https://cran.r-project.org) (electronic supplementary material, SI3 and figure SI3).

# 3. Results

## 3.1. Specimen description

The skull has been crushed laterally, obscuring the left side of the head and lower jaw, and remains hard to interpret despite high-resolution micro-CT scanning (see electronic supplementary material, S2 stored on Dryad for movies). Some of the skull bones in *Acherontiscus* are tightly knit, obscuring the sutures and suggesting that the animal was not a juvenile, although other bones are displaced. As preserved, the skull has a short snout with a relatively large, laterally placed orbit and a long postorbital region, but skull height and interorbital distance remain hard to estimate. The rear margin of the skull table is convexly curved and may consist of either one single or two large postparietals (figures 1 and 2), with a small tabular at each corner. The lacrimals and nasals are short, but the frontals are elongate and tapered anteriorly to be deeply wedged between the nasals. Other parts of the cheek and skull roof are difficult to separate. Lacrimals, nasals, premaxillae and maxillae bear lateral line pores, some of which are elongate (figure 2). Parts of the palate and braincase are preserved, showing the pterygoids with large but sparsely distributed denticles. The palatines bear small teeth, the ectopterygoid larger ones and the vomers do not appear to be preserved (figure 2). The parasphenoid is narrowly triangular, and the sphenethmoid is a broad V in cross section, appearing firmly attached to the skull roof, and bearing a double row of crests for attachment of the parasphenoid (figure 2). The dentary houses at least 18 teeth although the tip of the jaw is missing, the maxilla bears about 21 and the premaxilla 11 or 12. The pattern of a divergent number of upper versus lower teeth is not uncommon in early tetrapods, and of particular interest for our phylogenetic results, occurs in the colosteids [14] and the Tournaisian *Aytonerpeton* [4]. Substantial branchial bars are preserved, which are essentially similar to those of the contemporary adelogyrinids [5,10].

Micro-CT scans reveal the dentary tooth row to be dominated by four enlarged teeth at its centre, with smaller teeth anteriorly and posteriorly (figures 1 and 2). On the right side of the jaw, the bases of the large maxillary teeth and some of their tooth crowns have been broken through: none shows any sign of labyrinthodont infolding in the enamel. The tips of two of the enlarged teeth have penetrated the skull roof as the dentary was folded over by crushing and are laterally compressed with apicobasally ridged crowns and crenellated tips (figure 1). The maxilla has a similar number of enlarged teeth to the dentary (tips are not preserved) and although they are less enlarged than those on the dentary, they would have occluded with them (figures 1 and 2).

The lower jaw bears lateral line pores and has a surangular lateral line canal as well as a submandibular canal (figure 2). The probable third and second coronoids bear similar-sized denticles to the pterygoids (figure 2). The lower jaw also has a high surangular crest which would have supported the jaw-closing musculature and allowed a more powerful bite at the level of the enlarged dentition. A single Meckelian fenestra is present (figure 2). We can confirm the diplospondylous nature of the vertebral column as described by Carroll [9] but do not describe the postcranial skeleton further.

## 3.2. Phylogenetic analysis

No taxa were identified as being suitable for safe taxonomic deletion. The unweighted parsimony analysis yielded 312 filtered trees at 1237 steps, with an ensemble consistency index (C. I.) of 0.2753 (excluding uninformative characters) and ensemble retention index (R. I.) of 0.5699. Bootstrap and jackknife support were generally low for most nodes. Reweighting characters by the maximum value of their rescaled consistency index resulted in one tree (199.18768 steps; C. I. = 0.4542; R. I. = 0.7363). A single tree was retrieved after each implied weighting analysis, for any given value of the $K$ constant of concavity.

Regardless of optimality criteria and search settings, *Acherontiscus* is consistently retrieved as the sister taxon to the recently described Tournaisian tetrapod *Aytonerpeton* [4], and the (*Acherontiscus* + *Aytonerpeton*) clade forms the sister group to adelospondyls (*Adelogyrinus*, *Adelospondylus*, *Dolichopareias* [15]) (figure 3). However, nodal support for these groupings is invariably poor (electronic supplementary material, S1 and figure S3).

Under parsimony, the ((*Acherontiscus* + *Aytonerpeton*) + adelospondyls) clade emerges as sister group to a clade consisting of aïstopods (*Lethiscus* [7]; *Oestocephalus* [16]; and nectrideans (*Sauropleura*; *Ptyonius*; *Urocordylus* [15])). These appear in all analyses as sister group to a tetrapod array including, inter alia, colosteids and the Devonian *Tulerpeton* [17]. The implied weighting analysis resulted in few though significant changes in the position of some taxa, chiefly stem tetrapods. Among these,

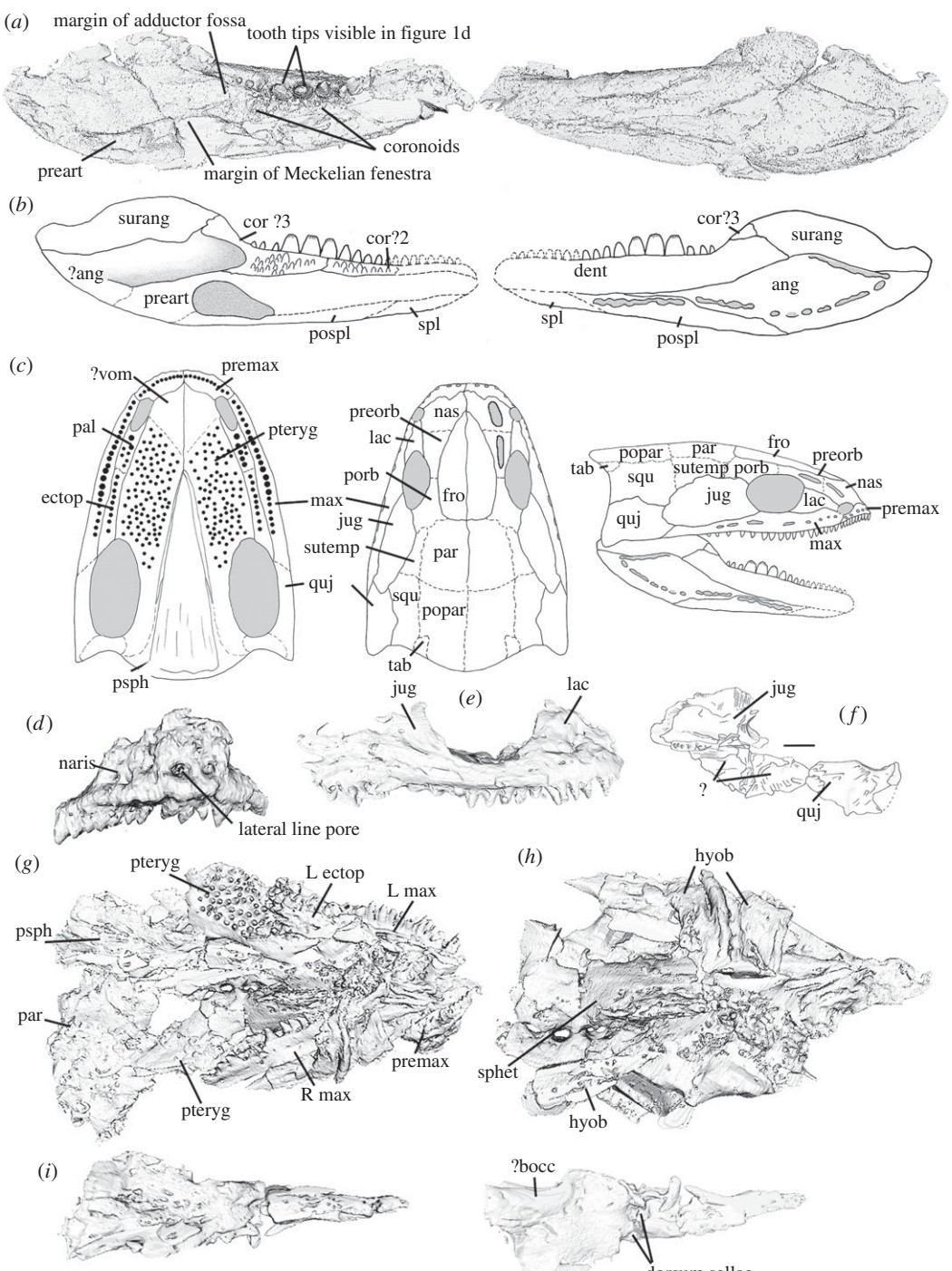

**Figure 2.** *Acherontiscus caledoniae* Micro-CT images and skull and lower jaw reconstructions. (*a*) Left lower jaw ramus from micro-CT scan, mesial surface at left, lateral surface at right. (*b*) Left lower jaw ramus reconstructions mesial surface at left, lateral surface at right. (*c*) Skull reconstructions, palate at left, skull roof centre, lateral view at right. (*d*) Left premaxilla. (*e*) Right maxilla, part of the jugal and other cirumorbital bones. (*f*) Camera lucida drawing of skull bones not visible in the scan. Scale bar 5 mm. (*g*) Micro-CT image of ventral surface of the skull with the lower jaw removed. (*h*) Micro-CT image of ventral surface of the skull with the lower jaw, marginal dentitigerous bones, and the parasphenoid removed. (*i*) Parasphenoid, ventral view at left, dorsal view at right. Figures except (*f*) not to scale. Abbreviation: ?bocc, ?basioccipital; ectop, ectopterygoid, hyob, hyobranchial elements; jug, jugal, L, left, lac, lacrimal; max, maxilla; par, parietal; preart, prearticular; premax, premaxilla; psph, parasphenoid; pteryg, pterygoid; quj, quadratojugal; R, right; sphet, sphenethmoid.

*Tulerpeton* appears in a more conventional position as the most derived Devonian stem-tetrapod, and a clade consisting of *Ossirarus* [4] and *Ossinodus* [18] branches between *Ventastega* [19] (anti-crownward) and *Ymeria* [20] (crownward) (figure 3).

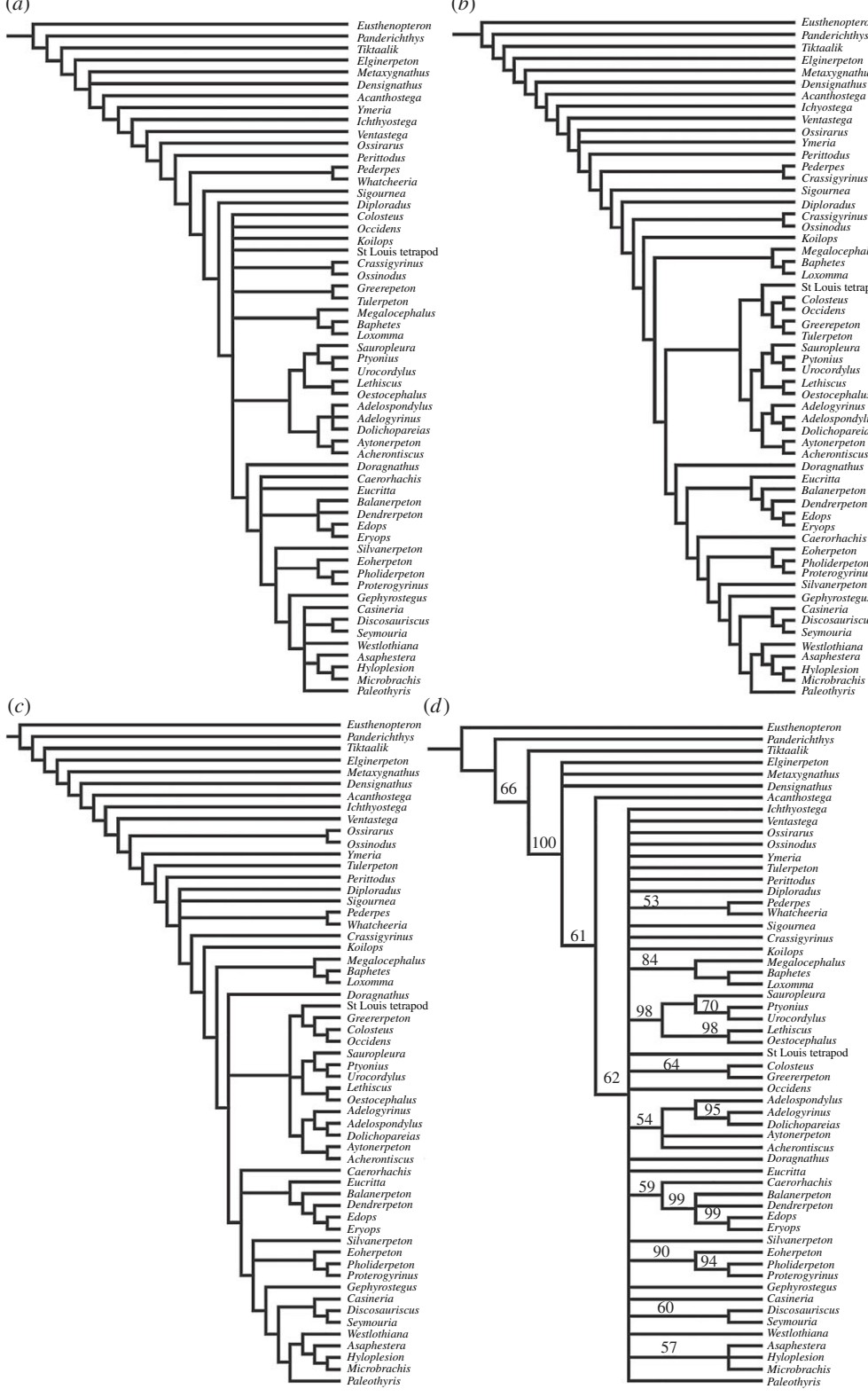

**Figure 3.** Interrelationships of major group of early tetrapods. (*a*) Strict consensus of 312 equally parsimonious trees with unweighted characters. (*b*) Single tree from parsimony analysis with characters reweighted by the maximum value of their rescaled consistency index. (*c*) Strict consensus of 10 trees, each obtained after applying implied weighting with a different integer value of the *K* constant of concavity (with *K* ranging from 1 to 10). (*d*) Bayesian topology with credibility values appended to tree branches.

In the Bayesian analysis, few clades emerge with moderate to strong support (Bayesian credibility values) and most taxa and groups are collapsed. Among the clades supported by the Bayesian analysis are (aïstopods + nectrideans) and ((*Acherontiscus* + *Aytonerpeton*) + adelospondyls) (figure 3).

# 4. Discussion

## 4.1. Heterodonty in non-mammalian tetrapods

The occurrence of heterodonty and durophagy in such an early tetrapod, and in particular, the form of the exposed tips of the enlarged dentary teeth is unprecedented. That these tips represent places where the infolded enamel has been removed by erosion to reveal the underlying dentine can be ruled out. Labyrinthodont infolding is usually (although not exclusively) associated with large teeth. Unfortunately, the resolution of the scan does not permit us to determine whether it was present in any of the smaller teeth, nor to see cross sections of the larger teeth. Furthermore, the dentary teeth are compressed laterally, whereas the skull is compressed dorsoventrally. The resolution of the scan is insufficient to show whether the tips of the smaller teeth are similarly shaped, or how thick the enamel may have been relative to the dentine. Enamel thickness significantly affects a tooth's function [21].

Among Palaeozoic fossil forms, a few Late Carboniferous and several Early Permian microsaurs show a degree of heterodonty (e.g. *Pantylus* [15]), but the most similar tooth distribution and morphology are seen in the Early Permian captorhinid eureptile *Opisthodontosaurus* from North America [22]. Formerly considered the earliest example of this kind of heterodonty, *Opisthodontosaurus* is more than 50 Myr younger than *Acherontiscus*. Furthermore, the skull of *Opisthodontosaurus* is about 2.5 times the size of that of *Acherontiscus*, and unlike the limbless *Acherontiscus* with its lateral lines, the captorhinid *Opisthodontosaurus* would have been terrestrial. The convergence in dentition, including the ribbed tips of the enlarged teeth, is therefore surprising. It may indicate an early example of convergence towards an effective crushing and slicing action powered by a strong bite force.

Heterodonty is closely tied to diet and occurs associated with omnivory, insectivory and herbivory [23]. A diet of arthropods with tough chitin was proposed for *Opisthodontosaurus* [22]. *Acherontiscus* might have fed on aquatic molluscs or crustaceans, including ostracods with their hard carapaces and which are abundant in the matrix from the fossil. The robust, high surangular crest associated with the derived dentition may indicate a new level of biomechanical complexity in the jaws of early tetrapods and might imply the presence of highly differentiated jaw adductor muscles, with a powerful bite.

The arthropod fauna of the mid-Carboniferous is poorly known, but as an aquatic tetrapod, and in view of its small size, *Acherontiscus* is unlikely to have had access to fully terrestrial forms such as myriapods, although it may have had access to smaller aquatic crustaceans.

Apicobasal ridges are not uncommon among tetrapods, for example, the Late Triassic rhynchocephalian *Eilenodon*, as well as various other extinct taxa [24]. In *Eilenodon*, it is thought to be associated mainly with herbivory, and especially consumption of lycopsid stems, enhancing tooth penetration with additional abrasive edges and greater grip. Patterns of heterodonty can appear similar in animals with similar diets, even across unrelated species [25,26]. Palynology implies that the environment from which *Acherontiscus* was recovered was a small body of still water surrounded by small herbaceous lycopsids. Larger *Lepidodendron* lycopsids formed a forest further away at other times (electronic supplementary material, S1). However, although lycopsids were very common in the likely habitat of *Acherontiscus*, there is no direct evidence that these formed one of its food sources. The first tetrapod herbivores are not thought to have evolved before the latest Carboniferous [27].

Heterodonty together with a durophagous morphology also occurs widely, but sporadically, among both later Palaeozoic and Mesozoic lineages. It appears in stem amniotes such as diadectids [28], and in crown amniotes such as the captorhinid *Opisthodontosaurus* [22], notosuchian crocodyliforms [29], various lineages of herbivorous dinosaurs [30], the sauropterygian placodonts [31] and Mesozoic and modern lepidosauromorphs [24,32,33]. In amphibians and their stem group, it is very rare: some early tetrapods including temnospondyls show size heterodonty and some fossil dissorophoid temnospondyls show bicuspid teeth, but none shows durophagy [34]; some modern caecilians have multicusped teeth [35]; and heterodonty is absent from anurans and urodeles.

Fishes (chondrichthyans, actinopterygian fishes, especially teleosts) and mammals are the groups that have been most studied for the developmental genetics of heterodonty (e.g. [36,37]), although there has been recent work on lizards [38,39]. However, dentitions, including heterodonty, appear to be engendered by transcription factors in the jaw that are conserved across all vertebrates [25,40]. Once gained, heterodonty is also apparently easily lost and can also be regained because the pattern of the genetic code is still present [40]. Heterodonty is essentially the norm in synapsids and throughout mammals and occurs apparently independently among many families of modern lizards [41]. Mammals differ from other tetrapods in their enamel construction, but in both cases, it is the differential deposition of hard tissue that is used to create details of their external form [26,42].

Heterodonty in early tetrapods could have been associated with the diversification of body form which they underwent in the mid-Carboniferous, exploiting new and varied food sources for which differently patterned teeth would have been required. The relationship between heterodonty and diet could have been driven by a process of correlated progression [43] between diet and dentitions as tetrapods explored different terrestrial and aquatic niches emerging during the Early and mid-Carboniferous, so the expectation might be that more should eventually be found in the fossil record.

## 4.2. Other palaeobiological inferences

Our results suggest that the evolution of limbless tetrapods with elongate postcrania occurred three times in the late Viséan and early Serpukovian: in aïstopods [15], in adelospondyls [15] and in *Acherontiscus*. Each has a different and specialized form of dentition, although what each was exploiting is unknown. Although these three are placed close together in our analyses, it appears that they developed their dental conditions and limblessness independently from a common limbed ancestor, whose descendents may also include nectrideans and/or colosteids. These groups also show some members with a tendency for trunk or tail elongation. The Bayesian analysis breaks apart former stem amniote taxa which are placed variously along the spine, with microsaurs as an independent group. All analyses place colosteids, nectrideans plus aïstopods and a clade containing *Acherontiscus*, *Aytonerpeton* and the adelospondyls all as stem tetrapods, remote from stem amphibians, stem amniotes and microsaurs, further supporting the hypothesis of the polyphyly of 'lepospondyls'.

One fauna contemporary with that of *Acherontiscus*, from the Burghlee Ironstone in the region of Loanhead, includes several taxa with specialized dentitions. It includes actinopterygians such as *Eurynotus* [44] and *Drydenius* [45] with durophagus adaptations, probably the enigmatic tetrapod *Caerorhachis* [46] with a totally denticulated lingual lower jaw surface and palate, a new small lungfish with an unusual durophagous dentition [47], and a range of taxa with numerous homodont chisel-shaped teeth (adelospondyls [5], *Doragnathus* [48]), and the broad, shallow-headed baphetid *Spathicephalus* [49]. All the tetrapod taxa except *Spathicephalus* appear unique to this area of Scotland, indicating an unusual set of conditions for the mid-Carboniferous.

In the mid-Carboniferous, the Loanhead area of Scotland was in the equatorial region of the Earth [50]. Today, the equatorial rainforest is home to the greatest diversity of life on Earth. The same is likely to have been true in the Carboniferous, providing the impetus for further diversification of tetrapods.

# 5. Conclusion

The enigmatic *Acherontiscus* plus the fauna from Loanhead exemplify and illustrate the expanded range of skull and body morphologies and dental organization among tetrapods that began to emerge in the mid-Carboniferous. This interval was key to the great diversification of tetrapods, which culminated in the evolution of amniotes in the early Late Carboniferous. Following the Hangenberg Event at the end of the Devonian, large plants initially disappeared, recovering slowly through the Tournaisian [4,51]. As dense floras and more complex ecosystems emerged throughout the Viséan, new niches became available in continental ecosystems, particularly in equatorial regions, and were exploited by tetrapods.

Consistent with new finds from the Tournaisian of tetrapods [1,4] and lungfishes [52], many vertebrate clades appear to have arisen much earlier than previously considered. Much must still be missing from the fossil record of the Early Carboniferous, and we might expect to find further examples of more specialized adaptations among tetrapods of that time. Continental deposits of the Early Carboniferous deserve further exploration and study.

Data accessibility. ESM 1-3 available on https://doi.org/10.5061/dryad.0pc151n. ESM 1 & 3 form a single pdf document. ESM S1 Palynological analysis: age and environment of *Acherontiscus*: ESM S3 Phylogenetic analysis, character list and data matrix. ESM S2 Micro-CT scan data and three movies. Files include: *Acherontiscus* scan data and 3-D files (3 movies). *Acherontiscus* movies: Skull with left lower jaw removed.mov; Left lower jaw.mov; Skull roof.mov.

Authors' contributions. A.R.M. and M.R. initiated the study, M.R. carried out phylogenetic analyses, J.A.C. and K.Z.S. contributed new micro-CT data, segmentation and skull reconstructions, J.E.A.M. contributed palynological dating and environmental analysis. J.A.C., M.R., A.R.M., T.R.S., K.Z.S. and J.E.A.M. contributed to writing the paper.

Competing interests. The authors declare there are no competing interests.

Funding. NERC consortium Grant (NE/J022713/1 Cambridge JAC, TRS, KZS: NE/J021091/1 Southampton, JEAM); Leverhulme Trust Emeritus Fellowship to J.A.C. A.R.M. received no funding.

Acknowledgements. Stig Walsh, for curation and loan of specimens, Andrew Ross, for advice on Viséan arthropods (National Museums Scotland: NE/J020621/1); Malcolm Burrows and José Casal-Jimenez (Cambridge Zoology) and

Javier Ortega-Hernández and Julie Sarmiento Ponce (Cambridge Earth Sciences) for microphotography; Farah Ahmed, Amin Garbout and Brett Clark (Natural History Museum, London [NHM]) for Zeiss Versa scanning; Marc Jones (NHM), Abigail Tucker (Kings College London) and Jason Head (University Museum of Zoology, Cambridge) for discussion of heterodonty in tetrapods.

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
