## [Reviewer comments · Royal Society Open Science]

Review History

RSOS-182087.R0 (Original submission)

Review form: Reviewer 1

Is the manuscript scientifically sound in its present form?

Yes

Are the interpretations and conclusions justified by the results?

Yes

Is the language acceptable?

Yes

Is it clear how to access all supporting data?

Yes

Do you have any ethical concerns with this paper?

No

Have you any concerns about statistical analyses in this paper?

No

Recommendation?

Accept with minor revision (please list in comments)

Comments to the Author(s)

Excellent job all around on getting these important new data from a difficult to interpret specimen. I would select "publish as is" but in the discussion of heterodonty the authors choose to highlight the later occurring (Permain) Pantylus but neglect the earlier (Westphalian) microsaurus from Jogjins and Nyrany that show some heterodonty. It doesn't really take away from the substance of the discussion but the timing would shave some time off between them and Acherontiscus. I am pleased to see the tetrapod stem group continues to gather these little guys and increased morphological disparity, but I think all would agree we have a lot more anatomy to describe and specimens to find before that part of the tree topology will stabilize.

Review form: Reviewer 2 (Robert Reisz)**Is the manuscript scientifically sound in its present form?**

Yes

Are the interpretations and conclusions justified by the results?

Yes

Is the language acceptable?

Yes

Is it clear how to access all supporting data?

Yes

Do you have any ethical concerns with this paper?

No

Have you any concerns about statistical analyses in this paper?

No

Recommendation?

Accept with minor revision (please list in comments)

Comments to the Author(s)

This is an important paper, well written and I support its publication. However, I do have some recommendations that I think would improve it.

In particular, they relate to the phylogenetic analysis section and the discussion. Also, some changes to the figures are recommended.

Phylogenetic analysis: How does inclusion of *Acherontiscus* in the Pardo et al. (2018) data matrix effect that tree topology? It is generally unhelpful if previous publications and analyses of early tetrapod phylogenies are largely ignored, as you do here. I would suggest that the issue of how *Acherontiscus* fits in the analysis of Pardo et al *Nature* paper should be discussed. The tree

topology of that analysis is very different from what is presented here, and I am wondering how does *Acherontiscus* fit into that tree, especially since the main topic of that paper is another limbless early tetrapod.

Discussion:

- 1) I highly recommend using the CT data for analyzing the dental morphology of the large teeth. Since it is not possible to do thin sections of the dentary teeth, I am strongly recommending visualizing the “durophagous” teeth in cross and vertical section using the tomographic data. Internal anatomy of the teeth should provide information about whether this represents teeth that are designed to break through hard tissues. If they have thick dentine, then you would have a very strong case for durophagy.
- 2) I agree that the combination of heterodonty and durophagy is rare in amphibians and their stem group, but size based heterodonty is known in trematopids, but of course no durophagous dentition. I would mention that.
- 3) Discussion from line 250-254 is a bit too simplistic. Heterodonty is widespread in early amniotes and stem amniotes, not just synapsids, as you well know. It is a widespread phenomenon among eureptiles, parareptiles, and even early diapsids (e.g. *Araeoscelis*) and beyond. So, I would restructure this portion of the discussion to include early non-synapsid amniotes and some microsaurians in this segment.

Figures and figure captions:

Figure 1 has two (c), please correct. Please identify/label the teeth in Fig. 1c (d).

I am also not in favor of the abbreviations used in this manuscript. The more commonly used codings are: f-frontal, l-lacrimal, m-maxilla, pmx-premaxilla, l-lacrimal, p-parietal, qj-quadratojugal, pt-pterygoid, etc. (this is the style of A.S. Romer and others for tetrapods and amniotes). Sorry, I am betraying my NA prejudice, but I think these figure labels are more compact and still informative.

What is lr, lower?? This is only part of labels used in figures and, and is therefore confusing. I personally find the use of -L lr jaw- inappropriate.

More importantly, if you can reconstruct completely the lower jaws and much of the skull, you should be able to label the constituent bones in figures 1 and 2. Same for the reconstructions. This is important for the more general reader, since only a handful of readers would know the difference between the coronoids, angular, surangular, etc. These are reconstructed but not labelled.

Decision letter (RSOS-182087.R0)

20-Feb-2019

Dear Professor Clack

On behalf of the Editors, I am pleased to inform you that your Manuscript RSOS-182087 entitled "Acherontiscus caledoniae, the earliest heterodont and durophagous tetrapod" has been accepted for publication in Royal Society Open Science subject to minor revision in accordance with the referee suggestions. Please find the referees' comments at the end of this email.

The reviewers and handling editors have recommended publication, but also suggest some minor

revisions to your manuscript. Therefore, I invite you to respond to the comments and revise your manuscript.

- Ethics statement

- Data accessibility

If you wish to submit your supporting data or code to Dryad (<http://datadryad.org/>), or modify your current submission to dryad, please use the following link:
<http://datadryad.org/submit?journalID=RSOS&manu=RSOS-182087>

- Competing interests

- Authors' contributions

- Acknowledgements

- Funding statement

Please ensure you have prepared your revision in accordance with the guidance at

<https://royalsociety.org/journals/authors/author-guidelines/> -- please note that we cannot publish your manuscript without the end statements. We have included a screenshot example of the end statements for reference. If you feel that a given heading is not relevant to your paper, please nevertheless include the heading and explicitly state that it is not relevant to your work.

Because the schedule for publication is very tight, it is a condition of publication that you submit the revised version of your manuscript before 01-Mar-2019. Please note that the revision deadline will expire at 00.00am on this date. If you do not think you will be able to meet this date please let me know immediately.

Please note that Royal Society Open Science charge article processing charges for all new

submissions that are accepted for publication. Charges will also apply to papers transferred to Royal Society Open Science from other Royal Society Publishing journals, as well as papers submitted as part of our collaboration with the Royal Society of Chemistry (<http://rsos.royalsocietypublishing.org/chemistry>).

on behalf of Dr Robert Sansom (Associate Editor) and Kevin Padian (Subject Editor)
openscience@royalsociety.org

Associate Editor Comments to Author (Dr Robert Sansom):

Associate Editor: 1

Comments to the Author:

Dear Prof Clack,

Thank you for your submission to RSOS. Both reviewers find the paper to be well written and support its publication and I am therefore very happy to recommend its publication subject to minor revisions. The requested revisions are very minor in the most part. Reviewer 2 queries the choice of phylogenetic matrix (and highlights Pardo et al 2018 as an alternative). I would suggest that in the revised manuscript that the taxon is added to that matrix as well as the one already analysed, or more simply still, a statement is added to the MS to justify your choice of your updated former matrix over that of Pardo et al 2018.

I look forward to receiving those revisions.

Rob

Reviewer comments to Author:

Reviewer: 1

Comments to the Author(s)

Excellent job all around on getting these important new data from a difficult to interpret specimen. I would select "publish as is" but in the discussion of heterodonty the authors choose to highlight the later occurring (Permain) Pantylus but neglect the earlier (Westphalian) microsaur from Joggins and Nyrany that show some heterodonty. It doesn't really take away from the substance of the discussion but the timing would shave some time off between them and

Acherontiscus. I am pleased to see the tetrapod stem group continues to gather these little guys and increased morphological disparity, but I think all would agree we have a lot more anatomy to describe and specimens to find before that part of the tree topology will stabilize.

Reviewer: 2

Comments to the Author(s)

This is an important paper, well written and I support its publication. However, I do have some recommendations that I think would improve it.

In particular, they relate to the phylogenetic analysis section and the discussion. Also, some changes to the figures are recommended.

Phylogenetic analysis: How does inclusion of *Acherontiscus* in the Pardo et al. (2018) data matrix effect that tree topology? It is generally unhelpful if previous publications and analyses of early tetrapod phylogenies are largely ignored, as you do here. I would suggest that the issue of how *Acherontiscus* fits in the analysis of Pardo et al *Nature* paper should be discussed. The tree topology of that analysis is very different from what is presented here, and I am wondering how does *Acherontiscus* fit into that tree, especially since the main topic of that paper is another limbless early tetrapod.

Discussion:

1) I highly recommend using the CT data for analyzing the dental morphology of the large teeth. Since it is not possible to do thin sections of the dentary teeth, I am strongly recommending visualizing the “durophagous” teeth in cross and vertical section using the tomographic data. Internal anatomy of the teeth should provide information about whether this represents teeth that are designed to break through hard tissues. If they have thick dentine, then you would have a very strong case for durophagy.

2) I agree that the combination of heterodonty and durophagy is rare in amphibians and their stem group, but size based heterodonty is known in trematopids, but of course no durophagous dentition. I would mention that.

3) Discussion from line 250-254 is a bit too simplistic. Heterodonty is widespread in early amniotes and stem amniotes, not just synapsids, as you well know. It is a widespread phenomenon among eureptiles, parareptiles, and even early diapsids (e.g. *Araeoscelis*) and beyond. So, I would restructure this portion of the discussion to include early non-synapsid amniotes and some microsaurs in this segment.

Figures and figure captions:

Figure 1 has two (c), please correct. Please identify/label the teeth in Fig. 1c (d).

I am also not in favor of the abbreviations used in this manuscript. The more commonly used codings are: f-frontal, l-lacrimal, m-maxilla, pmx-premaxilla, l-lacrimal, p-parietal, qj-quadratojugal, pt-pterygoid, etc. (this is the style of A.S. Romer and others for tetrapods and amniotes). Sorry, I am betraying my NA prejudice, but I think these figure labels are more compact and still informative.

What is lr, lower?? This is only part of labels used in figures and, and is therefore confusing. I personally find the use of -L lr jaw- inappropriate.

More importantly, if you can reconstruct completely the lower jaws and much of the skull, you should be able to label the constituent bones in figures 1 and 2. Same for the reconstructions. This is important for the more general reader, since only a handful of readers would know the

difference between the coronoids, angular, surangular, etc. These are reconstructed but not labelled.

Author's Response to Decision Letter for (RSOS-182087.R0)

See Appendix A.

Decision letter (RSOS-182087.R1)

01-Apr-2019

Dear Professor Clack,

I am pleased to inform you that your manuscript entitled "Acherontiscus caledoniae, the earliest heterodont and durophagous tetrapod" is now accepted for publication in Royal Society Open Science.

on behalf of Dr Robert Sansom (Associate Editor) and Professor Kevin Padian (Subject Editor)
openscience@royalsociety.org

Appendix A

Thank you for your submission to RSOS. Both reviewers find the paper to be well written and support its publication and I am therefore very happy to recommend its publication subject to minor revisions. The requested revisions are very minor in the most part. Reviewer 2 queries the choice of phylogenetic matrix (and highlights Pardo et al 2018 as an alternative). I would suggest that in the revised manuscript that the taxon is added to that matrix as well as the one already analysed, or more simply still, a statement is added to the MS to justify your choice of your updated former matrix over that of Pardo et al 2018.

I look forward to receiving those revisions.

Rob

Reviewer comments to Author:

Reviewer: 1

Comments to the Author(s)

Excellent job all around on getting these important new data from a difficult to interpret specimen. I would select "publish as is" but in the discussion of heterodonty the authors choose to highlight the later occurring (Permian) *Pantylus* but neglect the earlier (Westphalian) microsaur from Joggins and Nyranj that show some heterodonty. It doesn't really take away from the substance of the discussion but the timing would shave some time off between them and *Acherontiscus*. I am pleased to see the tetrapod stem group continues to gather these little guys and increased morphological disparity, but I think all would agree we have a lot more anatomy to describe and specimens to find before that part of the tree topology will stabilize.

Response: The time difference we mentioned refers specifically to the crown amniote *Opisthodontosaurus*, as being the most comparable type of heterodonty and durophagy. We have added 'Late Carboniferous' at line 201, to cover the microsaur.

Reviewer: 2

Comments to the Author(s)

This is an important paper, well written and I support its publication. However, I do have some recommendations that I think would improve it.

In particular, they relate to the phylogenetic analysis section and the discussion. Also, some changes to the figures are recommended.

Phylogenetic analysis: How does inclusion of *Acherontiscus* in the Pardo et al. (2018) data matrix effect that tree topology? It is generally unhelpful if previous publications and analyses of early tetrapod phylogenies are largely ignored, as you do here. I would suggest that the issue of how *Acherontiscus* fits in the analysis of Pardo et al *Nature* paper should be discussed. The tree topology of that analysis is very different from what is presented here, and I am wondering how does *Acherontiscus* fit into that tree, especially since the main topic of that paper is another limbless early tetrapod.

Response: We used our previous matrix rather than that of Pardo et al. for a number of reasons. First, our data includes new and unique data on a range of Tournaisian tetrapods that could not be included in Pardo et al.. These taxa have thrown up some novel arrangements of early tetrapod taxa (Clack et al. 2017). We further feel that our range of taxa, with fewer more derived forms and including more basal forms is the appropriate one for placing taxa from the Early Carboniferous. That of Pardo et al. seems to us to be overbalanced with those more derived taxa, including large groupings whose inter-relationships have been relatively stable over time. This set of taxa and characters swamps and rather undervalues the basal forms. We chose the range of taxa we felt better reflected that from the Carboniferous, especially the Early Carboniferous. It also includes more 'lepospondyls', whose inter-relationships remain controversial, and with which *Acherontiscus* has been previously associated.

Discussion:

1) I highly recommend using the CT data for analyzing the dental morphology of the large teeth. Since it is not possible to do thin sections of the dentary teeth, I am strongly recommending visualizing the “durophagous” teeth in cross and vertical section using the tomographic data. Internal anatomy of the teeth should provide information about whether this represents teeth that are designed to break through hard tissues. If they have thick dentine, then you would have a very strong case for durophagy.

Response: Unfortunately, although we had the specimen scanned in the highest resolution microCT machine available in the UK (the NHM Zeiss Versa), there was insufficient resolution to see cross sections of the teeth. We do state something about the resolution in the text, but I have added a phrase referring to the large teeth. Only a synchrotron scanner would give us sufficient resolution, and we do not have

resources or ready access to such a machine. We absolutely agree that it would be desirable to do this, and hope that some group will do this in the future. They may also be able to make more sense of the skull as a whole!

2) I agree that the combination of heterodonty and durophagy is rare in amphibians and their stem group, but size based heterodonty is known in trematopids, but of course no durophagous dentition. I would mention that.

Response: Trematopids and heterodonty. This person is not familiar with temnospondyls. *Phonerpeton* as figured by Dilkes has a pseudocanine, but similar disproportionately large marginal teeth can be found in the colosteid *Greererpeton*, the dvinosaurian *Erpetosaurus*, and clumps of pseudocanines can be found in *Cochleosaurus*, *Eryops* and many more. *Balanerpeton* had different sized lowers and uppers. This type of heterodonty is widespread.

Response: Done

3) Discussion from line 250-254 is a bit too simplistic. Heterodonty is widespread in early amniotes and stem amniotes, not just synapsids, as you well know. It is a widespread phenomenon among eureptiles, parareptiles, and even early diapsids (e.g. *Araeoscelis*) and beyond. So, I would restructure this portion of the discussion to include early non-synapsid amniotes and some microsaurians in this segment.

Response: Discussion of these groups occurs between lines 236 and 241. I have added trematopids on lines 242 and 243. I have mentioned 'microsaurians' at line 201-202 and changed 'recumbirostrans' for 'Late Carboniferous'. (See also for Referee 2)

Figures and figure captions:

Figure 1 has two (c), please correct. Please identify/label the teeth in Fig. 1c (d).

Response: Done

I am also not in favor of the abbreviations used in this manuscript. The more commonly used codings are: f-frontal, l-lacrimal, m-maxilla, pmx-premaxilla, l-lacrimal, p-parietal, qj-quadratojugal, pt-pterygoid, etc. (this is the style of A.S. Romer and others for tetrapods and amniotes). Sorry, I am betraying my NA prejudice, but I think these figure labels are more compact and still informative.

Response: I am afraid that I (JAC) absolutely disagree with this suggestion. I have used longer abbreviations for my entire career, as I think they are generally easier for readers to understand without have to go back to the abbreviations list. I seem to remember years ago that someone said to me that there was a standard usage for abbreviations, but I never found it to be the case among the papers I read. Indeed, consider this. When I first started my career, and in times previous to that, lettering on figures was manually added: by handwritten labels; by attached type-written lettering; or by 'Letraset' or its equivalent. Naturally, people needed the abbreviations to be as short as possible. Now that we have software to do the job for us, we can easily make them longer and more intuitive. If there is space, we needn't use abbreviations at all, as I have done in some places.

What is lr, lower?? This is only part of labels used in figures and, and is therefore confusing. I personally find the use of -L lr jaw- inappropriate.

Response: We are really not sure what the objection is. We have corrected one of the labels in Figure 1 which needed an added space. This objection is really just personal preference.

More importantly, if you can reconstruct completely the lower jaws and much of the skull, you should be able to label the constituent bones in figures 1 and 2. Same for the reconstructions. This is important for the more general reader, since only a handful of readers would know the difference between the coronoids, angular, surangular, etc. These are reconstructed but not labelled.

Done, with proviso added in the legend. I have not added further labels to Figure 1 apart from labelling the teeth in part c; otherwise I have labelled all the bones whose identity we can be sure of.